# Recurrent Falls as the Only Clinical Sign of Cortical–Subcortical Myoclonus: A Case Report

Giulia De Napoli [1,2], Jessica Rossi [3,4,*], Francesco Cavallieri [4], Matteo Pugnaghi [2], Romana Rizzi [4], Marco Russo [4], Federica Assenza [4], Giulia Di Rauso [3,4] and Franco Valzania [4]

1   Department of Biomedical, Metabolic and Neural Sciences, University of Modena and Reggio Emilia, 41125 Modena, Italy; giulia.denapoli1@gmail.com
2   Neurology Unit, OCB, Azienda Ospedaliero-Universitaria di Modena, 41126 Modena, Italy; matteopugnaghi79@gmail.com
3   Clinical and Experimental Medicine PhD Program, University of Modena and Reggio Emilia, 41125 Modena, Italy
4   Neurology Unit, Neuromotor & Rehabilitation Department, Azienda USL-IRCCS di Reggio Emilia, 42123 Reggio Emilia, Italy; cava_87@hotmail.it (F.C.); franco.valzania@ausl.re.it (F.V.)
*   Correspondence: jessicarossi.mail@gmail.com

**Abstract:** Some authors use the term cortical–subcortical myoclonus to identify a specific type of myoclonus, which differs from classical cortical myoclonus in that the abnormal neuronal activity spreads between the cortical and subcortical circuits, producing diffuse excitation. The EEG shows generalized spike-and-wave discharges that correlate with the myoclonic jerks. We report the case of a 79-year-old patient with a history of right thalamic deep hemorrhagic stroke, with favorable evolution. Fifteen years later, he was readmitted to the emergency department for episodes characterized by sudden falls without loss of consciousness. An EEG with EMG recording channel on the right deltoid muscle was performed, which documented frequent diffuse spike–wave and polyspike–wave discharges, temporally related to myoclonic jerks in the lower limbs. Brain MRI showed the persistence of a small right thalamic hemosiderin residue at the site of the previous hemorrhage. Antiseizure treatment with levetiracetam was started, with rapid clinical and electroencephalographic improvement. Our case may represent a lesion model of generalized epilepsy with myoclonic seizures. Furthermore, it highlights that lower limb myoclonus of cortical–subcortical origin may be an underestimate cause of gait disturbances and postural instability. Then, it may be reasonable to include the EEG in the diagnostic work-up of patients with recurrent falls.

**Keywords:** myoclonic epilepsy; thalamocortical circuit; stroke; thalamic hemorrhage; movement disorders



## 1. Introduction

Myoclonus is defined as a sudden, brief, jerky, involuntary movement of a muscle or a group of muscles, caused by an abnormal muscular activation (positive myoclonus) or an interruption of muscular activity (negative myoclonus) [1]. It can be found in a variety of conditions as a result of different pathophysiological mechanisms. Based on etiology, myoclonus is divided into four categories: physiologic, essential, epileptic, and symptomatic [2]. Healthy people can experience physiological jerky movements that produce minimal or no disability, such as sleep myoclonus or hiccups. Essential myoclonus is a disorder of unknown etiology in which myoclonus is isolated or the most prominent finding. Most commonly, myoclonus is associated with other symptoms and signs, and it is secondary to other neurological or medical conditions, such as neurodegenerative, drug-induced, toxic–metabolic, or inflammatory disease (symptomatic myoclonus). Finally, myoclonus can be part of a chronic seizure disorder (epileptic myoclonus). In this condition, myoclonus may occur as a component of a seizure, as the only manifestation of a seizure, or as one of several types of seizure within an epileptic syndrome [3].

Determining the etiology of myoclonus can be challenging and requires a multi-step approach, starting with history collection and physical examination [4]. Neurological examination (NE) allows the characterization of amplitude, anatomical distribution, and pattern of activation of the myoclonic movement, and it may reveal the presence of other concurrent neurological signs. Depending on its distribution over the body, myoclonus may be focal, multifocal, segmental, or generalized. The movements may occur spontaneously, during an action, or while maintaining a posture and can be triggered by tactile, acoustic, or visual stimuli (reflex or stimulus-sensitive myoclonus) [5,6]. Basic tests such as blood tests, neurophysiological studies, and brain imaging may be useful if the clinical history and physical examination are not sufficient to determine the etiology of myoclonus. Indeed, different clinical categories of myoclonus may share the same pathophysiological mechanism. Therefore, there is also a classification that considers the physiological origin of myoclonus, which can be cortical, cortico-subcortical, subcortical/non-segmental, segmental, or peripheral. Electrophysiological tests, including electroencephalography (EEG), multichannel surface electromyography (EMG), EEG-EMG polygraphy with back-averaging and evoked potentials, can help localize the source of myoclonic jerks along the neuraxis and differentiate between these physiologies [7,8].

Cortical myoclonus originates from an abnormal neuronal activity in the sensorimotor cortex. It is typically action-induced and stimulus-sensitive and mainly involves the face and the upper limb extremities, which are the body areas with the largest representation in the homunculi. The motor cortex is known to be particularly implicated in the control of fine movements rather than acting across multiple muscle segments. As a result, cortical myoclonus is most often multifocal or focal but may also have a segmental or generalized distribution [3]. Indeed, hyperexcitability can rapidly spread from an initial focus to other sensorimotor areas of the ipsilateral or contralateral cortex, resulting in synchronous activation of adjacent muscles or even bilateral muscles [9]. The main electrophysiological findings in cortical myoclonus are brief EMG discharges (25 ms to 100 ms duration), preceded by a focal EEG spike or sharp wave discharge with a short latency (<40 ms for arm) [10]. Often, the myoclonus EMG discharges occur in high-frequency rhythmic bursts or trains. If there is no evident cortical correlate on a basic EEG trace, a back-averaged EEG-EMG technique can help identify a pre-myoclonic potential [11]. The presence of giant cortical somatosensory evoked potentials (SEPs) and enhanced long-latency EMG responses (C reflex) to mixed nerve stimulation can further support the cortical origin of myoclonus [10].

The mechanism underlying cortical–subcortical myoclonus consists in abnormal excessive excitation of connections between cortical and subcortical structures (particularly the thalamus), which generates a diffuse activation of the motor areas [12,13]. This results in generalized paroxysmal jerks of the limbs that usually occur from rest. Due to diffuse cortical activation, other seizure phenomena can also be present in association with myoclonus. EEG generalized spike/polyspike and wave discharges that correlate with the myoclonic jerks are the hallmark of this condition. Myoclonus discharges on EMG have a duration of less than 100 ms. Cortical–subcortical myoclonus is typically found in some idiopathic generalized epilepsy (IGE), such as Juvenile myoclonic epilepsy (JME) and myoclonus with absence epilepsy.

The remaining categories of myoclonus (subcortical non-segmental, segmental, and peripheral) are characterized by a longer duration of EMG discharges and a lack of evidence of abnormal cortical excitability (no consistent EEG correlates), and they differ in the pattern of muscle activation. Subcortical non-segmental myoclonus can originate from different structures between the cortex and the spinal cord. In this case, the abnormal activity typically starts in a focal area and then spreads in both rostral and caudal directions, producing axial jerks. A multifocal pattern can also be observed. In contrast, in the segmental type, the discharges are confined to a few contiguous muscle segments [14,15].Lastly, peripheral myoclonus usually presents with irregular focal jerks due to a lesion in the nerve, plexus, or root [16].

Here we report the case of a patient who developed a clinical and neurophysiological picture resembling cortico-subcortical myoclonus about fifteen years after a right thalamic hemorrhagic stroke.

## 2. Case Presentation

A 77-year-old patient was admitted to the emergency department because of an episode of sudden fall without loss of consciousness. Dizziness and vertigo preceding the event were also reported. The most relevant element in his medical history was a right thalamic hemorrhage with a favorable outcome that occurred fifteen years before (Figure 1a). He had no family or personal history of epilepsy and no febrile seizures during childhood. An urgent brain CT scan was obtained without any evidence of recent ischemic or hemorrhagic lesions. The NE was normal, except for slight pronation of the right arm. A carotid/vertebral Doppler ultrasound, performed based on the hypothesis of a transient ischemic attack, was normal. After 24 h under medical observation without recording new clinical episodes, the patient was discharged home with antiplatelet therapy. Thereafter, he underwent a brain MRI (Figure 1b) that documented the persistence of a small right thalamic hemosiderin residue at the site of the previous hemorrhage, in the absence of new ischemic or hemorrhagic lesions. An EEG did not show evidence of epileptiform activity.

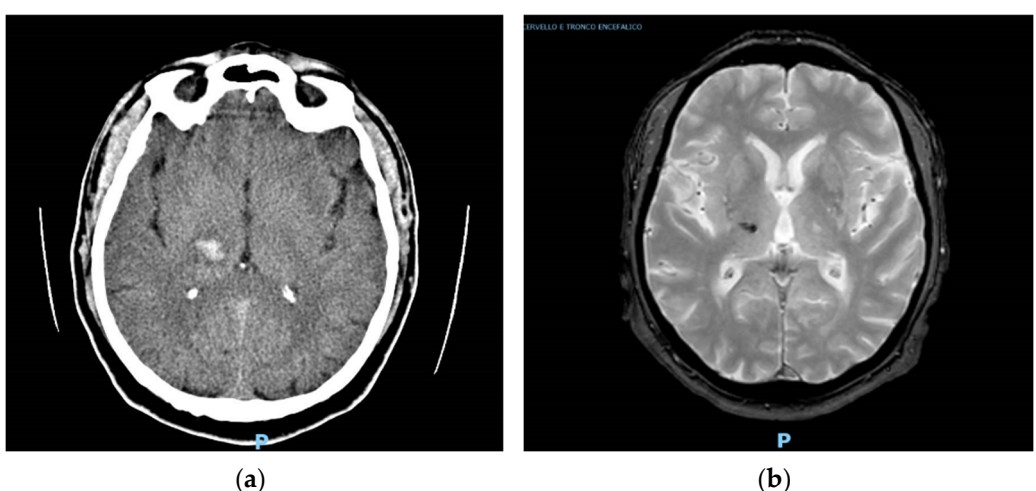

(**a**)                                      (**b**)

**Figure 1.** (**a**) Right thalamic hemorrhage detected on a CT axial scan fifteen years before the myoclonus onset; (**b**) T2-weighted MRI images performed after the first episode of myoclonus showing persistence of a small right thalamic hemosiderin residue at the site of the previous hemorrhage. "P" stands for "posterior": it indicates the back side of the image.

One year later, the patient was readmitted to the emergency department for recurrence of three episodes of sudden fall without loss of consciousness. NE revealed the presence of generalized jerks with major involvement of the lower limbs, occurring at rest but especially in maintaining antigravity postures. An EEG with EMG recording channel on the right deltoid muscle was performed in the emergency setting, which documented frequent diffuse spike–wave and polyspike–wave discharges, temporally related to myoclonic jerks (Figure 2). A new brain MRI was unchanged. Antiseizure treatment with levetiracetam up to 1000 mg/day was started, with rapid clinical and electroencephalographic improvement (Figure 3). After three years of follow-up, levetiracetam therapy was reduced to 500 mg/day in the absence of myoclonus recurrence.

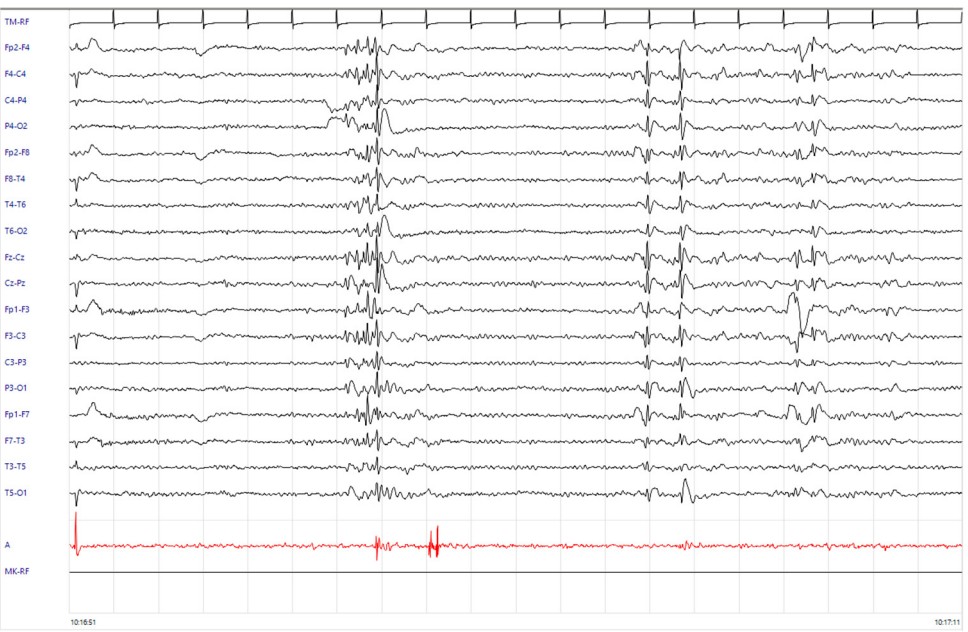

**Figure 2.** EEG with EMG recording channel on the right deltoid muscle, showing frequent diffuse spike–wave and polyspike–wave discharges, temporally related to the myoclonic jerks with a mean latency of 20 ms. Isolated generalized spike and wave discharges also occur without myoclonus as interictal phenomena.

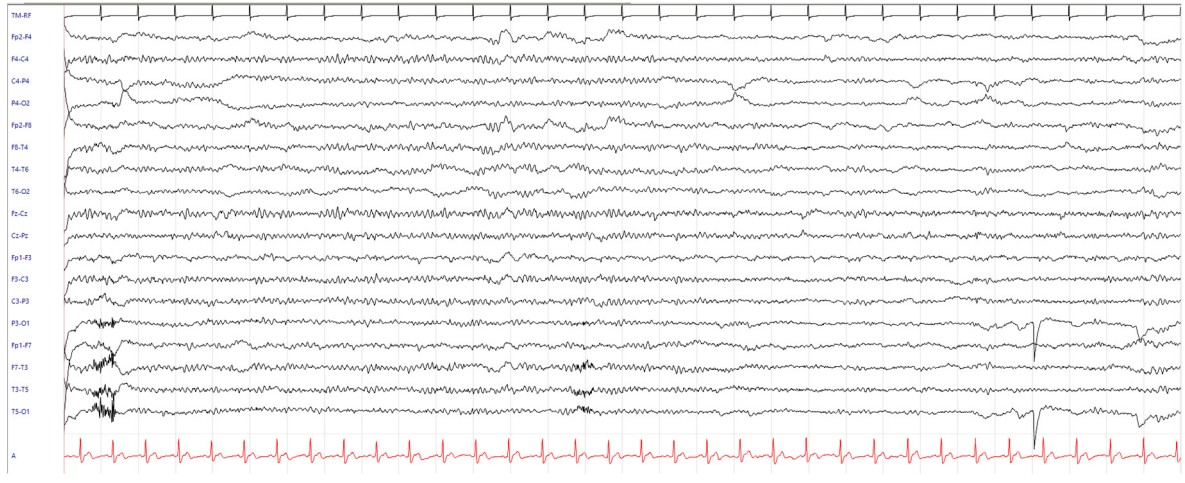

**Figure 3.** EEG recording after levetiracetam titration showing a normal trace without epileptiform activity.

## 3. Discussion

As displayed in Figure 2, our patient's EEG showed frequent diffuse spike–wave and polyspike–wave discharges, some of which were followed by a brief EMG discharge (80 ms) detected in the right deltoid muscle with a mean latency of 20 ms. These electrophysiologic findings are suggestive of a cortical–subcortical physiology, which is commonly seen in genetic generalized epilepsies, such as JME.

A growing body of evidence indicates that structural and functional alterations of the thalamocortical circuit are involved in the generation of generalized spike–wave discharges in IGE. Indeed, simultaneous EEG-fMRI acquisitions revealed an increase in thalamic neuronal activity along with widespread cortical deactivation during the generalized spike–wave discharge [17,18]. Furthermore, several quantitative MRI studies found gray matter alterations in the thalamus and in the frontal cortex, especially in the mesio-frontal area, in patients with JME as compared to controls [19]. In particular, a study conducted by

Kim et al. [20] showed a regional volume reduction in the anteromedial thalamus of JME patients with VBM analysis and a decreased functional connectivity between this thalamic region and the medial prefrontal cortex during resting-state fMRI.

In addition to the well-known thalamic involvement, many data suggest a possible role of basal ganglia in the modulation of generalized spike–waves in IGE. Specifically, simultaneous EEG-fMRI studies have shown hemodynamic changes in the striato-thalamo-cortical circuit prior to the generalized spike–waves in IGE patients [18,21]. Resting-state fMRI analyses found enhanced integration within the basal ganglia network in IGE [22]. Moreover, combined volumetric MRI and diffusion tensor imaging analyses revealed macrostructural and microstructural alterations in the left pallidum and bilateral putamen, hippocampus, and thalamus in patients with JME when compared to controls, thus supporting the pathophysiological hypothesis of striato-thalamo-frontal network abnormality underlying JME [23]. Rossi et al. recently reported the case of a patient with Wilson disease and a history of a seizure disorder resembling JME, who showed increased MRI functional connectivity within the motor system and between the globus pallidus and thalamo-frontal network [24]. According to the authors, these findings could be related to a possible role of basal ganglia damage due to the accumulation of ferromagnetic materials in the pathogenesis of the patient seizure phenotype. In the same way, our case may represent a lesion model of generalized epilepsy with myoclonic seizures. However, our patient presented a major involvement of the lower limbs, which is not the typical manifestation of myoclonic epilepsies and is more commonly found in myoclonus of subcortical origin. Therefore, despite the presence of a consistent EEG correlate, we can assume that the axial–subcortical component was more represented.

Movement disorders are a relatively uncommon complication of stroke, with an estimated prevalence of between 1 and 4% of all strokes [25]. They can manifest immediately after acute stroke or can develop later as a long-term sequela, sometimes with a progressive course. A review of 284 cases of post-stroke movement disorders (PSMDs) published between 1986 and 2016 showed a relatively scarce correlation between stroke location and type of movement disorder, thus reflecting the hypothesis that these abnormal movements derive from alterations of networks involving multiple regions of the brain [26]. After dystonia and chorea, myoclonus was the third most common PSMD and occurred equally in patients with hemorrhagic and ischemic strokes. It has been reported in association with thalamic stroke but also with ischemic lesions of the frontal cortex or the cerebellum. Likewise, depending on the region affected, thalamic stroke can produce a multitude of abnormal movements, with myoclonic manifestation predominating when the anterior or the antero-lateral region is involved. A systematic review on post-thalamic stroke movement disorders including 86 papers identified fifteen cases of myoclonus, reported as probably of subcortical type. The nuclei involved were centromedian, ventrolateral posterior, and ventroposterior [27]. To the best of our knowledge, no cases of cortico-subcortical myoclonus following a thalamic stroke have been reported.

Our patient developed myoclonus with a latency of 15 years after a hemorrhagic stroke in the postero-lateral region of the thalamus. The presence of a normal NE suggests selective involvement of inhibitory networks. The delay between the injury and onset of movement disorder may be attributed to mechanisms of neuronal plasticity [27]. During the first weeks after an ischemic or hemorrhagic injury, the motor system can reorganize itself to facilitate the recovery of the motor function through structural changes like axonal sprouting, dendritic remodeling, and synapse formation that take place around the affected area and also in distant brain regions [28]. However, maladaptive responses like diaschisis, dedifferentiation, and abnormal axonal regeneration may also occur, leading to the development of new and aberrant connections potentially manifesting as abnormal movements [26,28–30].

This study has several limitations, apart from being based on a single case. The detection of some patterns of muscular activation in the EMG channels could have provided additional useful information to better understand myoclonus characteristics. Unfortunately, since it was performed in an emergency setting, a complete polygraphic recording

was not available for our patient. Secondly, because of the good response to levetiracetam therapy, it was not deemed necessary to proceed with further investigations. Nevertheless, in our view, the clinical and electrophysiologic available data were sufficient to classify the myoclonus as of cortical–subcortical origin. Given the above, we assumed that the myoclonus could be attributed to functional and structural alterations of the thalamo-cortical networks resulting from aberrant regeneration mechanisms that were triggered by thalamic damage. Our case may thus represent a lesion model of generalized epilepsy with myoclonic seizures.

## 4. Conclusions

Our case suggests that a post-stroke lesion of the thalamus can generate a cortical–subcortical myoclonus, even several years after the insult has occurred. Moreover, lower limb myoclonus of cortical–subcortical origin may be an underestimated cause of gait disturbances and postural instability. Then, it may be reasonable to include the EEG in the diagnostic work-up of patients with recurrent falls, particularly if the clinical context could support a cortical–subcortical myoclonic origin.

**Author Contributions:** Conceptualization, J.R. and M.P.; investigation, J.R. and G.D.N.; writing—original draft preparation, G.D.N. and J.R.; writing—review and editing, G.D.N., J.R., F.C., M.P. and F.V.; visualization, G.D.N., J.R., F.C., M.P., R.R., M.R., F.A., G.D.R. and F.V.; supervision, J.R. and F.C.; project administration, J.R., F.C. and F.V. All authors have read and agreed to the published version of the manuscript.

**Funding:** This research received no external funding.

**Institutional Review Board Statement:** Not applicable.

**Informed Consent Statement:** Written informed consent has been obtained from the patient to publish this paper.

**Data Availability Statement:** Data sharing is not applicable to this article as no datasets were generated or analyzed during the current study.

**Conflicts of Interest:** The authors declare no conflicts of interest.

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
