# Peer review of "Recurrent Falls as the Only Clinical Sign of Cortical–Subcortical Myoclonus: A Case Report"

_neurosci, doi:10.3390/neurosci5010001_

Round 1
Reviewer 1 Report
Comments and Suggestions for Authors
The manuscript presented from De Napoli et al., entitled ‘Recurrent falls as the only clinical sign of cortical-subcortical myoclonus: a case report’ is an interesting case report. However it is not clear if this is the first report of a patient that presents the clinical sign described by the authors.
- The authors should extend the introduction to present this case report and they should improve the number of reference.
- The authors should improve the description of data analysis, it is not clear.
- It should be discussed the limit of this study/case.
Reviewer 2 Report
Comments and Suggestions for Authors
REVIEW
The manuscript presents a clinical case of a patient who exhibits clinical and neurophysiological signs related to cortico-subcortical myoclonus after fifteen years a right thalamic hemorrhage stroke. The authors suggest this case as a lesion model of generalized epilepsy with myoclonic seizures and the relevance to include EEG in the diagnostic of patients with recurrent falls
The introduction is brief, but considering this is a report of a specific clinical case and contains the basic information on the subject, it is acceptable.
The clinical case was correctly described and the discussion section provide enough argumentation and references to support the authors conclusions
Minor points
There is a “comment” in line 162 related to the reference [13]. Possibly, this is not the final version of the manuscript. The authors should check this issue.
Reviewer 3 Report
Comments and Suggestions for Authors
The present manuscript detail a case of a patient displaying recurrent falls most likely due to cortical-subcortical myoclonus, as suggested by EEG and EMG, with a positive response to levetiracetam. From the clinical context, the authors suggest to be due to right thalamic deep hemorrhagic stroke occurred 15 years later, and eventual lesion-derived abnormal regeneration in the thalamic-cortical network.
The case is well presented and the manuscript well written.
Round 2
Reviewer 1 Report
Comments and Suggestions for Authors
The authors improved the quality of the manuscript, it is now ready for publication.